# A simple strategy to enhance the speed of protein secondary structure prediction without sacrificing accuracy

**Sheng-Hung Juan**[1☯], **Teng-Ruei Chen**[1☯], **Wei-Cheng Lo**[1,2,3]*

1 Institute of Bioinformatics and Systems Biology, National Chiao Tung University, Hsinchu, Taiwan,
2 Department of Biological Science and Technology, National Chiao Tung University, Hsinchu, Taiwan,
3 The Center for Bioinformatics Research, National Chiao Tung University, Hsinchu, Taiwan

☯ These authors contributed equally to this work.
* WadeLo@nctu.edu.tw

**Data Availability Statement:** All relevant data are within the paper and its Supporting Information files.

**Funding:** This work was funded by the Ministry of Science and Technology (MOST), Taiwan (https://

## Abstract

The secondary structure prediction of proteins is a classic topic of computational structural biology with a variety of applications. During the past decade, the accuracy of prediction achieved by state-of-the-art algorithms has been >80%; meanwhile, the time cost of prediction increased rapidly because of the exponential growth of fundamental protein sequence data. Based on literature studies and preliminary observations on the relationships between the size/homology of the fundamental protein dataset and the speed/accuracy of predictions, we raised two hypotheses that might be helpful to determine the main influence factors of the efficiency of secondary structure prediction. Experimental results of size and homology reductions of the fundamental protein dataset supported those hypotheses. They revealed that shrinking the size of the dataset could substantially cut down the time cost of prediction with a slight decrease of accuracy, which could be increased on the contrary by homology reduction of the dataset. Moreover, the Shannon information entropy could be applied to explain how accuracy was influenced by the size and homology of the dataset. Based on these findings, we proposed that a proper combination of size and homology reductions of the protein dataset could speed up the secondary structure prediction while preserving the high accuracy of state-of-the-art algorithms. Testing the proposed strategy with the fundamental protein dataset of the year 2018 provided by the Universal Protein Resource, the speed of prediction was enhanced over 20 folds while all accuracy measures remained equivalently high. These findings are supposed helpful for improving the efficiency of researches and applications depending on the secondary structure prediction of proteins. To make future implementations of the proposed strategy easy, we have established a database of size and homology reduced protein datasets at http://10.life.nctu.edu.tw/UniRefNR.

www.most.gov.tw/?l=en) with grant number NSC 101-2311-B-009-006-MY2 to WCL. The funders had no role in study design, data collection and analysis, decision to publish, or preparation of the manuscript.

**Competing interests:** The authors have declared that no competing interests exist.

## Introduction

The secondary structure prediction (SSP) of a protein means to predict its per-residue backbone conformation merely based on the amino acid sequence. This technique has many applications, but there is an increasingly serious problem in its implementation, that is, the rapidly growing time cost. We believe that, if the speed of current SSP methods can be substantially enhanced, all fields relying on this technique will get benefits. This work thus aims to design a general strategy to cut down the time cost of SSP while preserving the accuracy for SSP methods. Currently, performing an accurate SSP for just one protein may take nearly an hour (see Results), greatly hampering large scale post-genomics applications. With the proposed strategy, a prediction with equivalent accuracy can be achieved in minutes.

Proteins are the basic functional units of biological systems. The function of a protein is dependent on its structure, which is determined by its amino acid sequence. Theoretically, as long as the structure of a protein is known, we will be able to identify or understand its biological functions. However, as genomes are sequenced much more rapidly than protein structures are solved, most proteins in biological databases nowadays have only sequence information but not structures. Therefore, in this post-genomics era, sequence-based protein structure prediction is of particular importance because by doing that, we may further be able to predict the function of a protein and then achieve many applications. The SSP is a subcategory of protein structure prediction and a key step toward the prediction of the tertiary structure. If the efficiency of SSP can be enhanced, many research and application fields will also be improved, such as the multiple sequence alignment and homology detection for proteins [1–3], the identification of disease-causing genetic mutations or variations [4–6], and the prediction of enzyme target sites [7,8], binding sites [9], antibody epitopes [10–12], protein-protein interactions [13,14] and protein subcellular localizations [15,16].

SSP has been developed for more than 65 years [17], during which the predictive feature set has evolved several times as the algorithms advances and the power of computers increases. In the 1970s, amino acid propensities and residue physiochemical properties were used to perform SSP based on statistical approaches [18]. Soon after that, the concept of "window" enabled statistical analyses of the interaction between features of adjacent residues and achieved ~60% prediction accuracy [19]. As machine learning techniques were applied since the late 1980s, the scale of the feature set has also been expanded. Take the PHD [20], a neural network method, for instance: it used multiple sequence alignment profiles as features to accomplish a three-state (helices, strands, and coils) SSP accuracy of ~70%. The Psipred, another machine learning SSP method, first utilized the position-specific scoring matrix (PSSM) generated by the PSI-BLAST [21] to be the feature set and pushed the three-state accuracy to 76.5% in 1999 [22]. From then on, the PSSM became the major feature set of SSP. Most highly accurate methods nowadays used it, inclusive of the RaptorX [23], SpineX [24], SSpro8 [25], Scorpion [26], Spider2 [27], and DeepCNF [28]. The three-state (Q3) accuracy of these methods was about 80–82%. The SSpro8, Scorpion, and DeepCNF were able to perform the eight-state (Q8) prediction, and the accuracy was around 67–72%.

A PSSM is generated by retrieving homologs of a query sequence from a target dataset and then, based on the alignment of these homologous sequences, computing the normalized substitution rates of 20 amino acids and transforming these rates into logarithm scores for each residue. This technique was first applied by Gribskov *et al.* to the identification of distantly related proteins [29]. With the refinements by 1) the Henikoffs using a weighting scheme to reduce the influence of sequence redundancy [30], 2) Tatusov *et al.* using a psuedocount algorithm to overcome the insufficiency of homologs from small target datasets [31], and 3) Altschul *et al.* integrating several protein similarity comparison algorithms into the

PSI-BLAST [21], the PSSM became increasingly robust in fields relying on protein sequence analyses, including the SSP.

A target dataset is essential for generating a PSSM. In the SSP field, since the big jump of Q3 accuracy made by Psipred, the target dataset utilized by most methods has been the UniRef90 maintained by the UniProt (Universal Protein Resource) [22–28]. Because of the rapid developments in next-generation sequencing technologies and genomics, the size of the UniRef90 dataset has increased dramatically recently. In Feb. 2019, there have been over 90.0 million proteins in this dataset, around 1.7 times the size of two years ago, Feb. 2017 (52.6 million). Although the UniRef90 is truly a great target dataset for SSP, as we keep using this rapidly growing dataset, the disk storage, memory space and computation time required to perform SSP will all dramatically increase as well. Therefore, we would like to examine whether it is necessary to use such a huge dataset, or perhaps there can be a feasible strategy for shrinking the size of the target dataset while preserving the accuracy. We noticed that the size of UniRef90 had increased exponentially in 10 years, but the Q3 accuracy of state-of-the-art SSP methods had stayed at a plateau of 80–82% [17]. Besides, we discovered that the sequence homology of the target dataset would greatly influence the number of zero entries in the probability matrix of a PSSM (see S1 File). Based on these observations, we hypothesized that 1) the size of target dataset would have a greater effect on the time cost than on the accuracy of SSP, and 2) the sequence homology level of the target dataset might affect the quality of the generated PSSM, which would then influence the accuracy of SSP.

By random sampling, the target dataset can be shrunk. We created many shrunk versions of the UniRef90 of 2015. Performance evaluations of seven state-of-the-art SSP methods supported our hypothesis. Even when UniRef90 was reduced to 1/16 in size and time cost was dramatically cut down by 93.1%, the average accuracy of those methods only decreased by 1.2%. By reducing the sequence redundancy of the target dataset, the dataset can also be shrunk, and, based on our hypothesis, the accuracy might be changed. Very interestingly, homology reduction experiments revealed that the accuracy was not only changed but improved. When examining how the accuracy was improved, we discovered that the Shannon information entropy might be capable of measuring the quality of a PSSM and explaining the accuracy it produces.

Although the accuracy may be decreased because of target dataset size shrinkage, it can be improved by homology reduction. We thus proposed a balanced strategy that a homology reduction of the target dataset to 25% sequence identity accompanied with a size shrinkage to 5 million proteins by random sampling will greatly cut down the time cost while preserving the accuracy of an SSP method. Finally, we evaluated this strategy with state-of-the-art SSP methods using the UniRef90 of 2018. The high accuracy was maintained while the speed was enhanced by 20.9 folds. We hope that this strategy can be widely adopted by current protein secondary structure prediction systems, the performance improvement of which shall be advantageous to researches and applications where SSP plays a key role, especially in this post-genomics era.

To sum up, this study aims to find the main factors that influence the speed of SSP and design a strategy to accelerate it. The experimental results not only supported our hypotheses but also helped us discover a shortcut to achieve a remarkable enhancement of the speed while preserving the high accuracy.

## Results

### Effects of target dataset size on the performance of SSP

We hypothesized that the influence of the target dataset size on the time cost of SSP would be much more significant than on accuracy. Based on this hypothesis, it would be expected that if

the target dataset were shrunk, the reduction of computation time would be much more apparent than the decrease in accuracy. In this experiment, we shrank the UniRef90-2015 dataset by random sampling and created subsets with $1/2^k$ size, where $k$ ranged from 0 to 20. The actual sizes of these subsets decreased from 38.2 million, 19.1 million . . ., to 37 proteins. Seven state-of-the-art SSP algorithms were tested with the TS115 dataset (see [17] and Materials and Methods) as the query set. This experiment was repeated ten times to obtain the average and standard deviation of the performance measures. As demonstrated in Fig 1, all methods showed the same trends. The time cost reduced rapidly as the target dataset size was shrunk, and the difference in time cost between the largest and smallest target datasets could be 3355 folds. On the contrary, except for SSpro8 [25], the decrease of accuracy was negligible for large target datasets, and the maximum difference in accuracy, regardless of the three- or eight-state measures, was less than 14.1%.

All the assessed methods utilized PSI-BLAST as the sequence alignment search and PSSM generation engine. A finer examination of the SSpro8 method let us recognize that the E-value threshold settings of PSI-BLAST in the SSpro8 pipeline script were very different from other methods. The E-value thresholds of the PSSM generation and alignment search stages set by SSpro8 were $10^{-10}$ and 0.001, respectively, whereas the other six methods set these thresholds as $10^{-3}$ and 10, respectively. Lower E-value thresholds would make PSI-BLAST discard more statistically insignificant homologs and speed up the generation of PSSM and dataset search; however, they would also sacrifice the quality of PSSM and the precision of search results [32,33], which might lead to the decrease in SSP accuracy. Indeed, after we modified the E-value settings of SSpro8 to be the same with other methods, despite the increase of time cost, all accuracy measures were greatly improved to the equivalent level of others (compare Fig 1G and 1H). This refined setting was thus applied to SSpro8 throughout this study. See S1 Table for the settings of PSI-BLAST for all assessed SSP methods.

The results of these methods were averaged and summarized in Fig 2 and S2 Table. The time cost decreased linearly as the target dataset size reduced (note that the horizontal axis is in logarithm). On average, the decrease of accuracy was minimal as long as the target dataset was larger than 1/8 of the UniRef90-2015. The decrease became obvious as the target dataset size was between 1/16 and $1/2^{14}$ of the UniRef90-2015; for target datasets smaller than this range, the decrease of accuracy slowed down. These results supported our hypothesis, and we found that the curves of time cost (TC) and accuracy of these size reductions could be fitted by the following equations,

$$\text{TC}(n_t) = 2.8658 \times 10^{-5} \cdot n_t + 5.8248 \tag{1}$$

$$\text{Q3}(n_t) = \frac{0.1363}{(1 + \text{Exp}(\frac{17.0963 - \text{Log}_2(n_t)}{2.1447}))^{0.6205}} + 0.6716 \tag{2}$$

$$\text{Q8}(n_t) = \frac{0.1281}{1 + \text{Exp}(\frac{15.6095 - \text{Log}_2(n_t)}{2.5405})} + 0.5621 \tag{3}$$

where $n_t$ stands for the size (number of proteins) of the target dataset. As illustrated in Fig 2B, these equations agreed well with the experimental results. It is noteworthy that, although the trends demonstrated by these equations might be tangible, the constants associated with them only fitted the TS115 query set, a PSSM target dataset with <90% identities (*e.g.*, the UniRef90), and our hardware/software setup (3.33 GHz CPU + 166 GB RAM + single PSI-BLAST thread; see S1 Table and Materials and Methods).

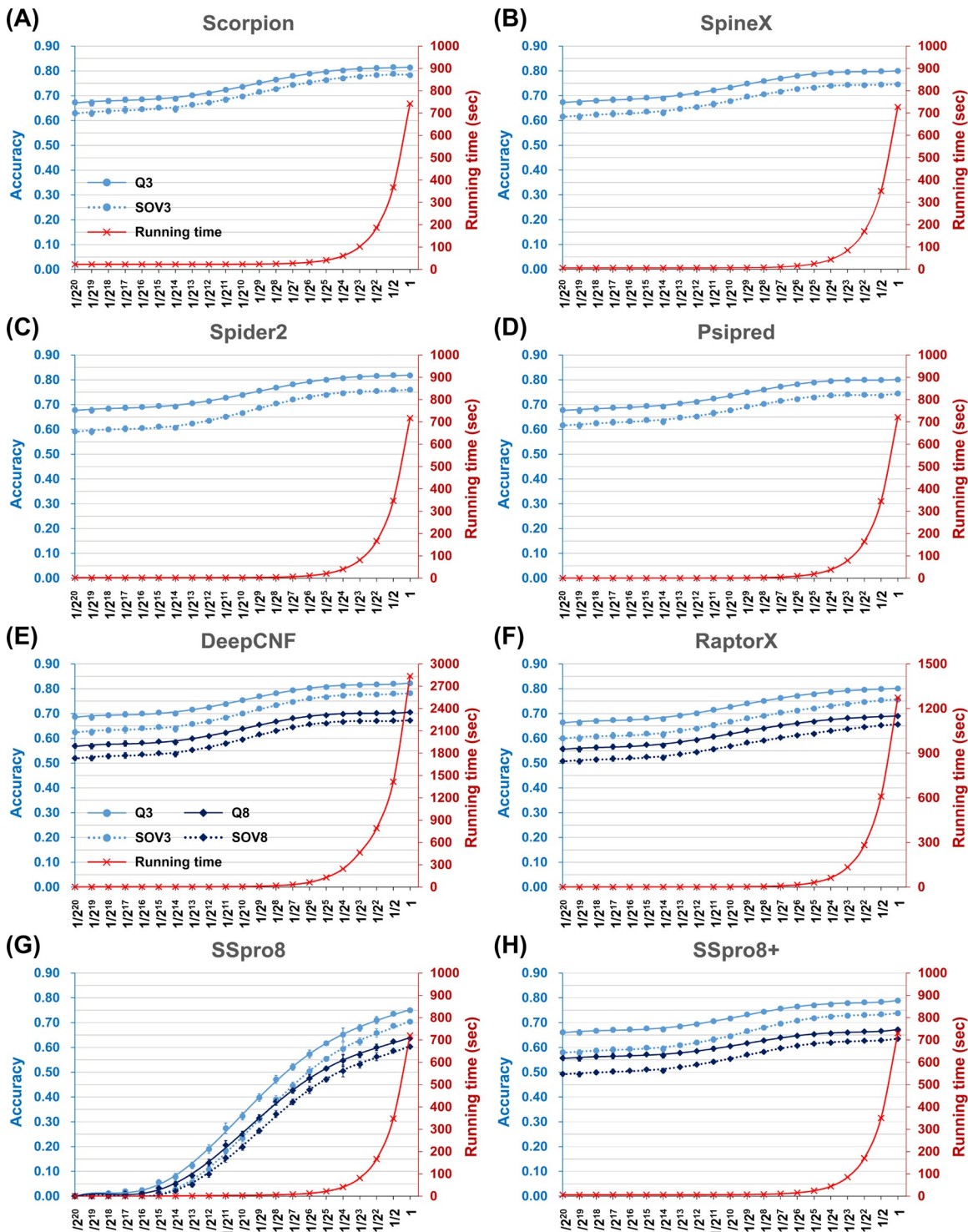

**Fig 1. The time cost and accuracy of state-of-the-art secondary structure prediction methods assessed with PSSM target datasets of decreasing size.** (**A**) Scorpion. (**B**) SpineX. (**C**) Spider2. (**D**) Psipred. (**E**) DeepCNF. (**F**) RaptorX. (**G**) SSpro8, template-free mode. (**H**) SSpro8 with refined PSI-BLAST settings, template-free mode. The query dataset used in this experiment was the TS115 prepared by [17], and the PSSM target sequences were all sampled from UniRef90-2015. The horizontal axis is the extent of target dataset size reduction, where 1 represents the original size of the UniRef90-2015. The vertical axes respectively indicate the time cost (red) and accuracy (blue). The computed accuracy measures include Q3, Q8, SOV3, and SOV8. For clarity, the color codes are displayed only in (A) and (E). All these state-of-the-art SSP methods exhibited the same tend as the target dataset size was reduced. The decrease was much quicker in time cost than in accuracy. To speed up prediction, the SSpro8 adapted a very low E-value setting for the PSSM generator PSI-BLAST;

however, we found by setting this, the accuracy was much sacrificed (G). After the setting was modified to be the same with other methods (see S1 Table), the accuracy of SSpro8 was much preserved (H). Using the full-sized UniRef90-2015, for any 3-state predictors, the average time cost for one query protein was ~12 minutes. As for the 8-state predictors, the time costs varied a lot. The DeepCNF was the most accurate algorithm; it took 47 minutes to predict one protein. The experiment was repeated ten times, and standard deviations are plotted on the curves, the deviations of time cost may be too small to see, though (refer to S2 Table for the raw data).

To examine whether the trends of these equations were robust when independent query sets were applied, we performed the same experiment on the CASP12 and CASP13 query sets. Just like the TS115, these datasets were composed of novel protein structures deposited in the PDB (Protein Data Bank) [34] after the assessed SSP methods were published. As summarized in S1 Fig and S2 Table, the trends of time cost and accuracy obtained based on the results of CASP12 and CASP13 agreed well with Eqs (1)–(3). The relationship between the time cost and the target dataset size was linear, while that between the accuracy and the target dataset size was sigmoid. For example, the equations of the time cost were,

$$\mathrm{TC}_{\mathrm{CASP12}}(n_t) = 2.8980 \times 10^{-5} \cdot n_t + 6.4610 \tag{4}$$

$$\mathrm{TC}_{\mathrm{CASP13}}(n_t) = 3.2020 \times 10^{-5} \cdot n_t + 5.5190 \tag{5}$$

where they were both in the form of a classic linear function ($y = \mathrm{m}x + \mathrm{b}$). The slope (or, the m) of CASP13 was larger than TS115 and CASP12 because CASP13 contained more large proteins than the latter two. Because TS115 contains many more proteins than the CASP datasets, in this report, we use the results of TS115 as the mainstream to explain our discoveries and make those of the CASPs available in the Supporting information files. The conclusions of this study are all applicable to the three query sets.

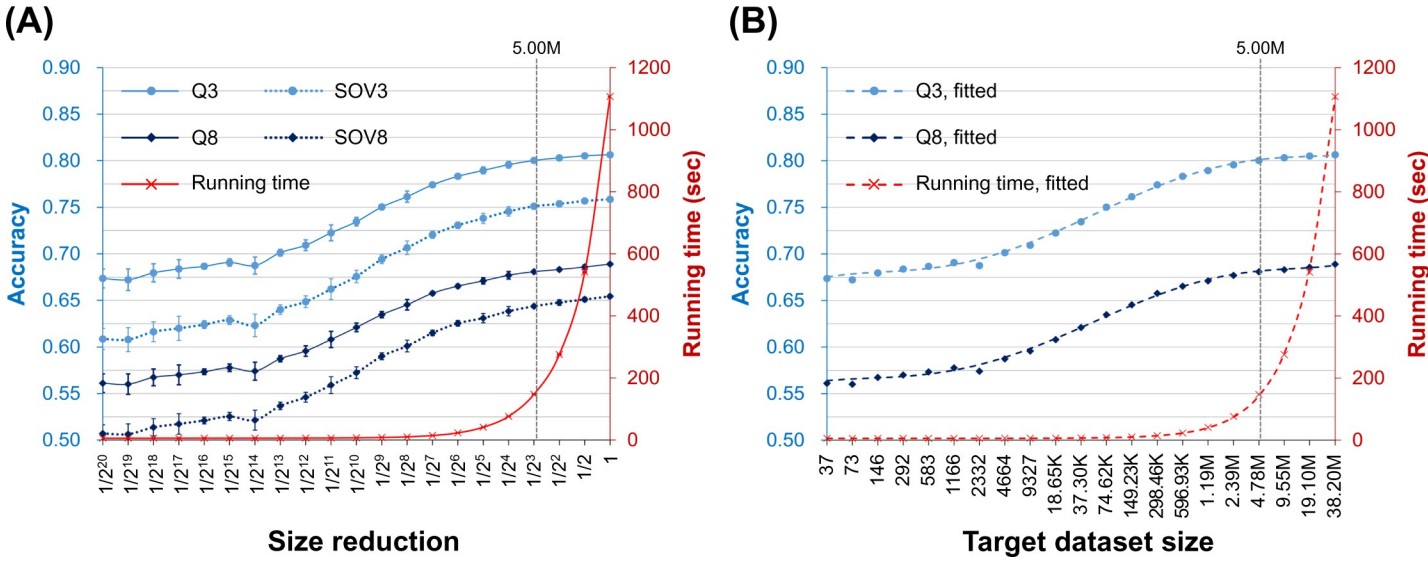

**Fig 2. The average time cost and accuracy of the assessed secondary structure prediction methods as the target dataset size reduces.** (A) The curves of actual experimental results averaged over seven SSP methods. The meanings of the axes of this plot are the same as those of Fig 1, but the scale of the vertical axis of accuracy (blue) is adjusted to start from 0.5 to make visible the standard deviations, which were obtained by ten repeats of random sampling test for each assessed method. Supporting our hypothesis, the decrease in time cost was much faster than that in accuracy. Provided that the target dataset sampled from the UniRef90-2015 was larger than ~5 million sequences (indicated by the dotted vertical line), the decrease in accuracy was minor. (B) The curves of fitted Q3 and Q8 accuracies. The horizontal axis indicates the number of proteins in the PSSM target dataset (M: million, K: thousand). The points of accuracies are drawn by the actual data, but the curves are made by Eqs (2) and (3), showing that those equations fit well with the experimental results.

## Effects of target dataset sequence redundancy on the performance of SSP

In addition to sampling, another way to shrink a target set is homology reduction, that is, making sequence identity non-redundant (NR) subsets. In general, a homology-reduced subset of lower identity contains fewer sequences. Since the target dataset size decreases as the sequence redundancy lowers, based on our first hypothesis and the results of the first experiment, it can be sure that the time cost of SSP will be cut down. As for whether the accuracy will decrease as the redundancy lowers, the situation is more complicated.

The second hypothesis of this study says that the homology level of the target dataset would affect the quality of PSSM and thus the accuracy of SSP. Accordingly, it would be expected that as the sequence redundancy decreases, the accuracy might diverge from the curves of Eqs (2) and (3), which consider only the dataset size. The question is: Will the accuracy be higher or lower than the curves? Since we had observed that a PSSM of low complexity might produce decreased accuracy and that the raw probability values of a PSSM generated from homologous sequences of high redundancy are usually less complicated than those generated from non-redundant sequences (S1 File), we supposed that homology reduction of the target dataset might improve SSP accuracy.

Using UniRef90-2015 as the source of target sequences, we created a series of homology reduced target datasets to perform this experiment (Materials and Methods). The sequence identity of these non-redundant datasets ranged from 80% to 25%. Fig 3 shows the average time cost and accuracy of the seven state-of-the-art SSP algorithms tested with TS115 as the query dataset (see S2 Fig and S3 Table for the plots and raw data of individual algorithms). In this figure, the curves made based on Eqs (1)–(3) were provided too. Agreed again with our first hypothesis, even when the target dataset was shrunk because of homology reduction, the time cost dropped. More interestingly, the Q3 and Q8 both went higher than the curves of Eqs (2) and (3), not only supporting our second hypothesis but implying that a low homology helps improve the accuracy. The CASP12 and CASP13 query datasets were also applied to perform this experiment, and the conclusion obtained from them was consistent with the above (refer to S3 Fig for the average performance and S3 Table for the raw data).

To understand how the homology of the target dataset influenced SSP accuracy, we tried to quantify the complexity of the PSSM by applying the Shannon information entropy [35] widely used to measure the disorder of data of a variable. A higher information entropy computed from a PSSM stands for a higher disorder of the probability matrix, which we suppose to reflect a higher complexity of the PSSM. As shown in Fig 4, where the entropy and Q3 accuracy of target datasets with decreasing homology are drawn in one plot, the two variables seemed to have some correlation. The Pearson's correlation coefficient between them was 0.417, a positive relationship. Similarly, the correlations between them obtained from the CASP12 and CASP13 tests were positive (S4 Fig). However, because as we reduced the homology the target dataset size was passively shrunk as well (meaning that there were two changing factors in this experiment), the actual correlation between entropy and accuracy should be carefully reexamined by fixing one of the two factors.

## Effects of sequence reduction of target dataset with a fixed dataset size on the performance of SSP

Here we selected a fixed dataset size to dissect further the influence of target sequence homology on the performance of SSP. Referring to Fig 2 and Eqs (2) and (3), we found that the decrease of accuracy was negligible as long as the target dataset was larger than 1/8 of the Uni-Ref90-2015, *e.g.*, ~5 million sequences. Thus, we decided to use this number of proteins as the fixed dataset size to repeat the previous experiment. Because this size was smaller than the NR

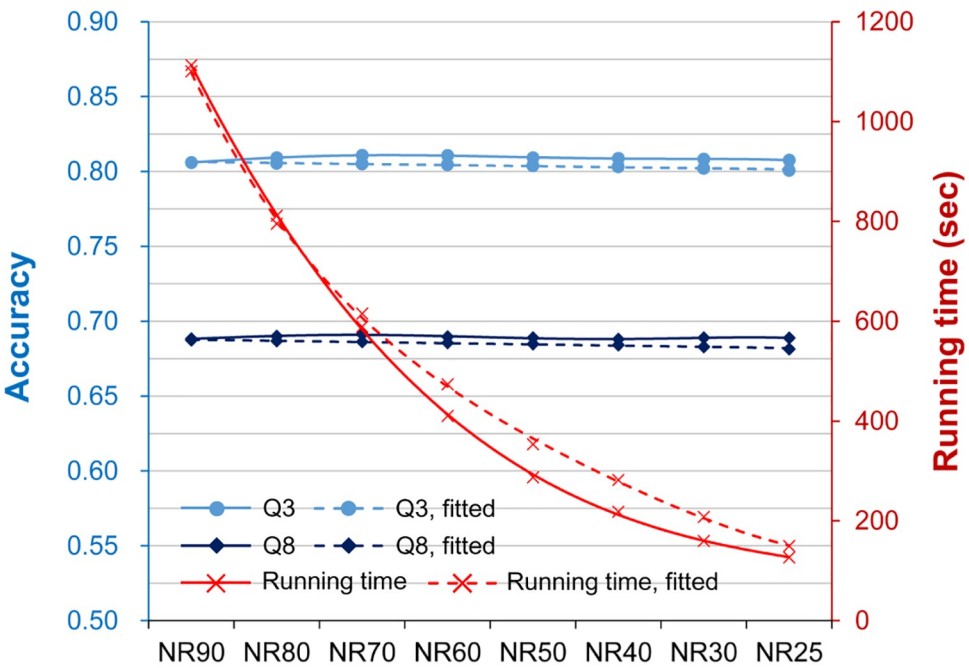

**Fig 3. The average time cost and accuracy of the assessed secondary structure prediction methods as the sequence homology of the target dataset reduces.** The horizontal axis is the extent of homology reduction in sequence identity non-redundancy. For example, NR90 means that any two sequences in the PSSM target dataset share <90% identities. The lowest sequence identity was 25% because when a lower identity such as 20% was applied, the number of sequences remaining in the target dataset would be far smaller than the 5 million safety zone (see Fig 2 and S6 Table) and then cause unreasonable decreases of the accuracy. The right vertical axis is the time cost (red), and the left is the accuracy (blue). The dotted red curve plots the expected time cost according to the Eq (1) formulated in the experiment of target size reduction. Because as the homology of the target dataset was reduced the dataset was also significantly shrunk, the time cost of SSP methods decreases rapidly. The dotted light and dark blue curves were the expected values of Q3 and Q8 made according to the size-reduction accuracy Eqs (2) and (3), respectively, while the solid blue curves showed the actual values. Very interestingly, as the target dataset homology reduced, the accuracy increased slightly.

datasets of the previous experiment, multiple repeats by random sampling could be applied. According to our hypotheses and the results of the above experiments, when the TS115 was applied as the query dataset, we expected that the time costs of these size-reduced NR datasets would all be similar while the accuracy would go higher than the curves of Eqs (2) and (3).

The results summarized in Fig 5 demonstrate that, when the target dataset size was fixed at 5 million proteins, the average time cost of the assessed methods stayed steady with a slight decrease as the homology reduced from 90% to 25% identity (see S5 Fig for the plots of particular algorithms). Importantly, the accuracy values were all higher than the values depicted according to Eqs (2) and (3). Although the average differences in Q3 and Q8 between the 90% and 25% NR datasets were both only 0.70%, the average $p$-values were $<1.25\times10^{-6}$ and $<9.99\times10^{-8}$, respectively (see S4 Table for the raw data and $p$-value of each method), indicating that these improvements in accuracy were statistically significant. Moreover, this improvement compensated for the decrease in accuracy caused by the shrinkage of the target dataset. Comparing S2 and S4 Tables, the Q3 achieved by the 5-million-protein target dataset of 25% identity was 0.1% higher than the Q3 achieved by the full UniRef90-2015 on average. As for the average Q8, the value accomplished by the former dataset was only 0.1% lower than that

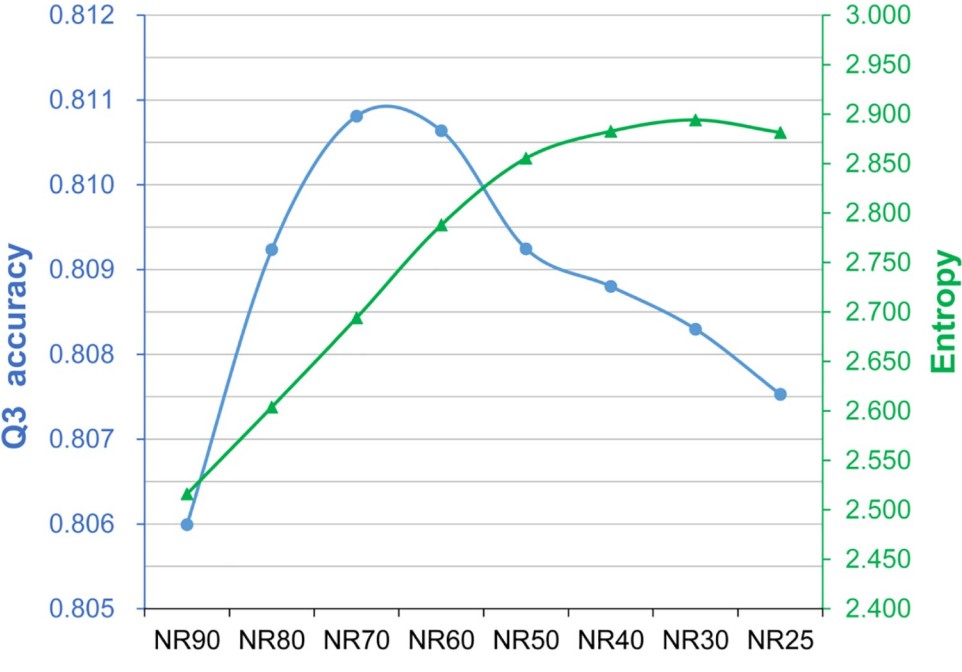

**Fig 4. The average information entropy and Q3 accuracy of the assessed secondary structure prediction methods as the sequence homology of the target dataset reduces.** The horizontal axis is the extent of target dataset homology reduction. The right vertical axis is the Shannon information entropy (green), and the left is the Q3 accuracy (blue). These two curves showed similar trends. The Pearson's correlation coefficient was 0.417, a positive relationship.

accomplished by the latter. The CASP query sets were also tested. On average, the Q3 and Q8 of them obtained with the UniRef25-2015 of 5 million proteins were merely 0.3‰ and 0.4% lower than those obtained with the full UniRef90-2015, respectively (S6 Fig and S4 Table). According to these data and those shown in Fig 3, we concluded that homology reduction of the target dataset could improve SSP accuracy. Even if the accuracy might be lowered by size reduction of the target dataset, the homology reduction could counteract the effects and restore/improve the accuracy.

We also computed the information entropy of the PSSM generated from these size-fixed NR target datasets (Fig 6). Compared with Fig 4, the trend of entropy got much more similar to Q3 as the homology of the target set decreased. The Pearson's correlation coefficient between them was 0.983, a strong relationship. The correlation coefficients between entropy and Q8, SOV3, and SOV8 were 0.947, 0.970, and 0.917, respectively. Consistent with these TS115 results, the correlation coefficients between the entropy and Q3 obtained with the CASP12 and CASP13 were 0.966 and 0.948, respectively, and the coefficients between the entropy and other accuracy measures all indicated strong positive correlations (>0.863; see S7 Fig). These results suggested that the information entropy of a PSSM may help explain the SSP accuracy it produces.

## The proposed strategy for the speed enhancement of SSP–with performance assessments

We discovered that, although shrinking the target dataset would decrease the accuracy, as long as the dataset is large enough, the decrease is minimal. Meanwhile, given a fixed target dataset size, homology reduction would increase the accuracy, especially at low identity levels. Taken

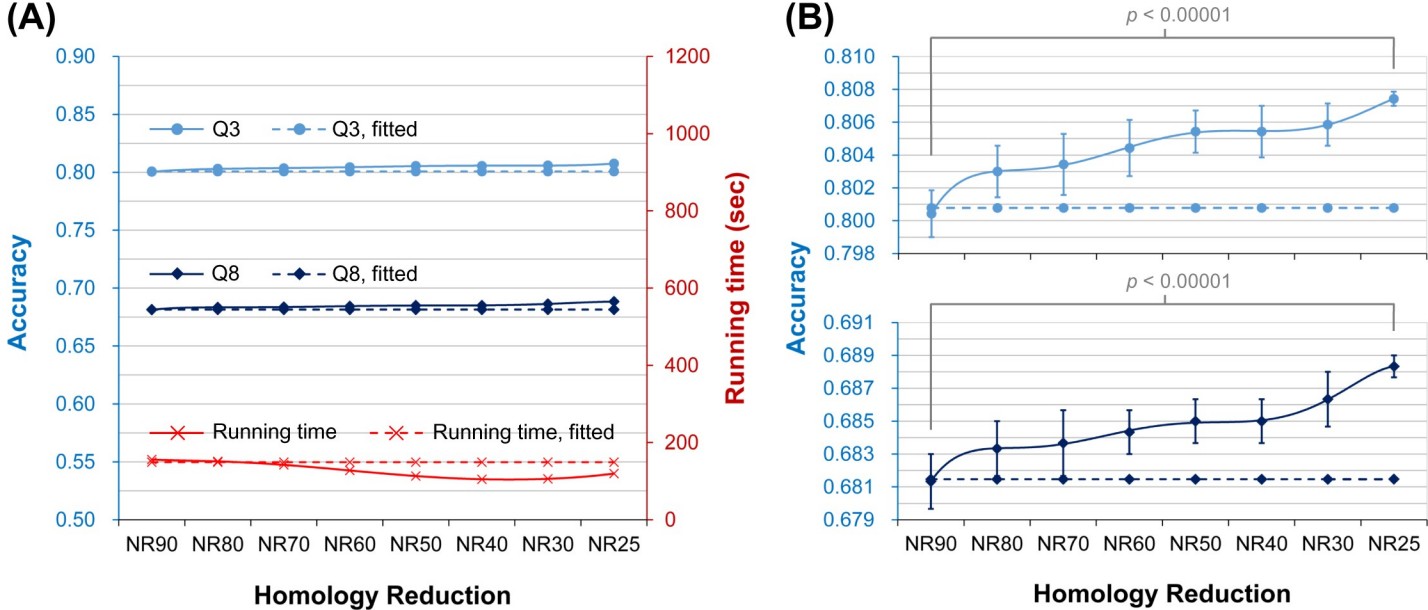

**Fig 5. The average time cost and accuracy of the assessed secondary structure prediction methods, as the sequence homology of the target dataset reduces with a fixed dataset size.** (A) The results of time cost, Q3, and Q8. (B) The results of Q3 and Q8, with the axis of accuracy zoomed in to make visible the standard deviations. In both plots, the dotted lines indicate the expected time cost or accuracy, according to the Eqs (1)–(3) formulated by target dataset size reduction. The size of these target datasets was fixed as 5 million proteins. The actual time costs (the solid red curve) by the datasets of low homology were slightly lower than the predicted, indicating that homology reduction of the target dataset can slightly improve the speed of SSP. More interestingly, the accuracies achieved by low homology sets were not only higher than the values predicted based on Eqs (2) and (3) but also significantly higher than the accuracies achieved by high homology sets. These data were averaged over seven methods. All $p$-values of these methods for the difference in Q3 or Q8 between the NR25 and NR90 datasets were $<10^{-5}$. To conclude, homology reduction of the PSSM target dataset helps **improve** SSP accuracy.

together, here we proposed a strategy to accelerate SSP without sacrificing the accuracy, that is, using a size and homology both reduced UniRef dataset to serve as the PSSM target dataset instead of the full UniRef90. The recommended extent of reduction is 5 million proteins with <25% sequence identities. If for some studies the accuracy is much more critical than the speed, an alternative strategy is simply just the homology reduction of the target dataset. According to the results demonstrated in Fig 3, it is expected that, if the size of the target dataset is much larger than 5 million proteins, a homology reduction at 70% sequence identity is sufficient to enhance the accuracy.

All the above experiments were performed using target sequences released no later than 2015. To test whether the proposed strategy is promising for future applications, we assessed it with the UniRef90 of the year 2018 as the source of target sequences. We expected that 1) by using Eqs (1)–(3), the time cost and accuracy of the full UniRef90-2018 could be well predicted, 2) the time cost could be significantly cut down, and 3) the accuracy achieved by the size-homology-reduced target dataset, the UniRef**25**-2018, would be close to that by the full UniRef90-2018. As shown in Table 1, the equation-predicted time cost and accuracy for the full UniRef90-2018 agreed well with the actual results. By reducing the size of the target dataset to 5 million proteins, the speed was increased by 20.9 folds. Importantly, all the accuracy measures achieved by the size-homology-reduced UniRef**25**-2018 were close to those by the enormous full UniRef90-2018. As for the alternative strategy, when only a 70% identity homology reduction was applied such that the target dataset was passively shrunk to 46.2 million proteins (UniRef70-2018), in addition to a 2.1-fold speed improvement, the three-state and eight-state accuracies were enhanced by 0.5–0.8% and 0.1–0.2%, respectively. See S5 Table for detailed data and the results of particular SSP methods.

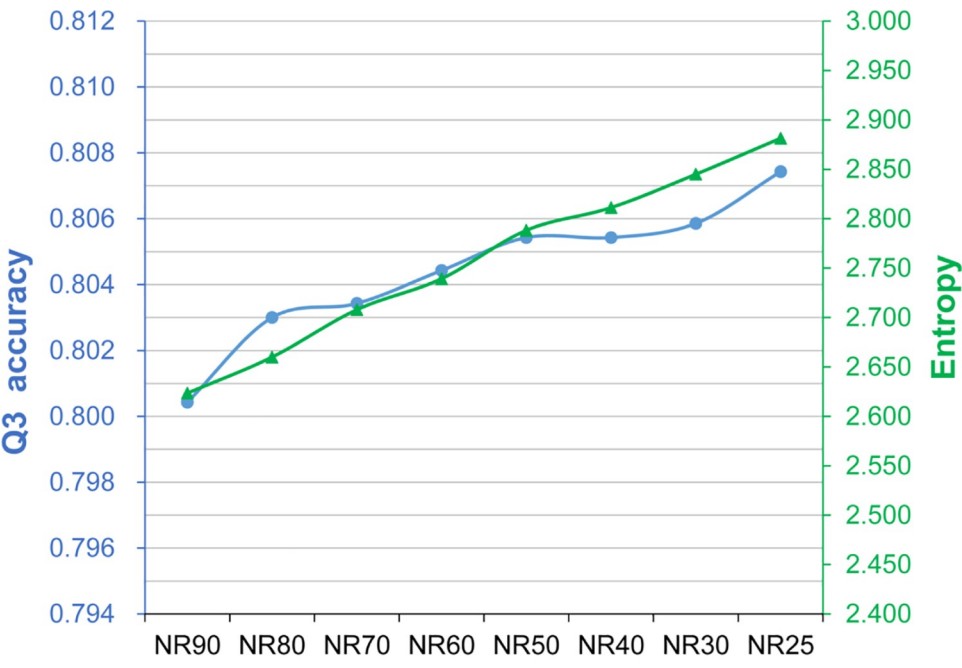

**Fig 6. The average information entropy and Q3 of the assessed secondary structure prediction methods, as the homology of the target dataset reduces with a fixed dataset size.** The horizontal axis is the extent of homology reduction. The color codes are the same as Fig 4. When the target dataset size was fixed at 5 million sequences, *i.e.*, the boundary of the safety zone (Fig 2), the PSSM information entropy and SSP accuracy showed a strong correlation, implying that the entropy may be helpful to explain how the quality of a PSSM influences the accuracy of SSP.

To examine the feasibility of the proposed strategy thoroughly, we now tested it oppositely by using an extremely large PSSM target dataset with sequence homology higher than 90% identity to assess the performance of the state-of-the-art SSP methods. The non-redundant protein dataset maintained by the NCBI (National Center for Biotechnology Information)

**Table 1. Predicted and actual performances of state-of-the-art SSP methods assessed with the original and manipulated UniRef 2018 target datasets.**

| Target dataset | Full UniRef90-2018 (87.3 million proteins) | | Full UniRef70-2018 (46.2 million proteins) | Sampled UniRef25-2018 (5.0 million proteins) |
|---|---|---|---|---|
| | Predicted[a] | Actual | | |
| Running time (sec) | 2507.575 | 2438.778[b] | 1181.185[c] | 116.938 ± 0.890[b,c] |
| Q3 | 0.807 | 0.806 | 0.811 | 0.804 ± 0.002 |
| SOV3 | 0.759 | 0.756 | 0.764 | 0.755 ± 0.003 |
| Q8 | 0.688 | 0.690 | 0.691 | 0.683 ± 0.002 |
| SOV8 | 0.655 | 0.655 | 0.657 | 0.646 ± 0.003 |

[a]The query set of this experiment was the TS115. The running time, or time cost, is predicted by Eq (1). The Q3 and Q8 were predicted by Eqs (2) and (3), respectively. The predicted SOV values were computed by fitting the curves of SOV shown in Fig 2A.

[b]All these performance measures were averaged over seven SSP methods, inclusive of this time cost. On average, using the full UniRef90-2018, it took 41 minutes to predict one protein. Accurate methods required a longer time. For instance, the most accurate method DeepCNF took 93 minutes for one protein.

[c]If predicted by Eq (1), the time costs of the full UniRef70-2018 and the sampled 5-million-protein UniRef25-2018 datasets would be 1330 and 149 sec, respectively. The actual time costs of them were only ~90% and ~80% of the predicted values, respectively, demonstrating again that homology reduction of the target dataset also slightly improves the speed of SSP.

collected the most comprehensive protein sequence data [36] and was also utilized by some modern SSP methods as the default target dataset, such as the Psipred [22]. At the time of this article, the NCBI NR dataset contained 257.1 million sequences sharing <100% identities. In order to predict the time cost and accuracy for this NrNCBI100-2020 dataset, the results of homology reduction experiments on the UniRef 2015 and 2018 obtained with the TS115 query set were combined (see S6 Table) to fit the curves shown below,

$$TC_h(n_t) = -4.0386E{-}14 \cdot n_t{}^2 + 3.2338E{-}05 \cdot n_t - 73.9573 \tag{6}$$

$$Q3_h(c_t) = -1.0552E{-}7{\cdot}c_t{}^3 + 1.4931E{-}5 \cdot c_t{}^2 - 0.0006c_t + 0.8152 \tag{7}$$

$$Q8_h(c_t) = -9.3552E{-}8{\cdot}c_t{}^3 + 1.5515E{-}5 \cdot c_t{}^2 - 0.0008c_t + 0.7000 \tag{8}$$

where $TC_h$, $Q3_h$, and $Q8_h$ stand for, respectively, the time cost, Q3, and Q8 accuracy of the homology-reduced target dataset, $n_t$ denotes the number of target sequences, and $c_t$ means the sequence identity cutoff of the target dataset.

These equations are very different from Eqs (1)–(3) because, in this experiment, the homology and size of the target dataset may both change. According to these equations, when the state-of-the-art SSP methods were performed with TS115 as the query set and NrNCBI100-2020 as the target dataset, the average time cost, Q3, and Q8 would be 5,571 sec, 0.799, and 0.682, respectively, where $n_t$ = 257.1 million (proteins) and $c_t$ = 100 (% identity). In other words, it was expected that the time cost of using NrNCBI100-2020 would be much higher than using UniRef90-2015/2018 because of its large size and that the accuracy obtained with NrNCBI**100** would be lower than the UniRef**90** datasets because of the increased sequence redundancy. After the real test, the time cost, Q3, and Q8 were 5,756 sec, 0.789, and 0.676, respectively (see S7 Table for the raw data of particular methods), very close to the expected values. These results supported that using a low homology target dataset with a suitably-shrunk size is the right direction for improving the efficiency of SSP.

## Discussion

### On the speed enhancement by size reduction of the target dataset

The results of all experiments in this study support our first hypothesis that the size of the target dataset exerts a much greater influence on the time cost of SSP than accuracy. Although the average time cost of the assessed SSP algorithms could be well fitted by Eq (1), there were some factors not considered yet. The pipeline of SSP methods nowadays typically comprise two stages, 1) the generation of PSSM by PSI-BLAST sequence similarity search against the target dataset, and 2) the prediction by machine learning with the PSSM as predictive features. The computation time is the sum of the two. A comprehensive analysis of the computation time of the assessed methods reveals that the former is the major part. As exemplified in Table 2 (see S1 Table for full data), the PSSM generation time of all methods is linearly proportional to the size of the target dataset, meaning time complexity $T \propto O(n_t)$ where $n_t$ is the number of proteins in the dataset, agreed with [32]. The time cost of prediction, however, almost remains constant regardless of the dataset size, except for DeepCNF [28]. The main reason why the prediction time of DeepCNF is much longer than that of other algorithms and rises as the target dataset enlarges is that it conducts another round of sequence similarity search against the target dataset in the prediction stage [28].

Because the majority of SSP algorithms spend most of the computation time in generating PSSM, any strategy capable of reducing the PSSM generation time will speed up the whole

**Table 2. The PSSM generation and secondary structure prediction time of the assessed SSP methods.**

| Method | | Target dataset size (number of proteins) | | | | | | | | |
|---|---|---|---|---|---|---|---|---|---|---|
| | | **0.15M** | **0.30M** | **0.60M** | **1.19M** | **2.39M** | **4.78M** | **9.55M** | **19.10M** | **38.20M** |
| Scorpion | PSSM | 2.27 | 4.56 | 9.22 | 18.52 | 38.11 | 79.30 | 164.01 | 344.55 | 719.39 |
| | Pred | 22.76 | 22.83 | 22.85 | 22.72 | 22.75 | 22.81 | 22.79 | 22.83 | 22.90 |
| SpineX | PSSM | 2.28 | 4.55 | 9.11 | 18.39 | 37.94 | 79.41 | 164.00 | 344.74 | 720.54 |
| | Pred | 5.94 | 5.93 | 5.93 | 5.95 | 5.96 | 5.95 | 5.98 | 5.98 | 5.99 |
| Spider2 | PSSM | 2.84 | 5.22 | 9.87 | 19.17 | 38.75 | 80.35 | 164.58 | 424.24 | 914.54 |
| | Pred | 1.34 | 1.17 | 1.06 | 0.95 | 0.93 | 0.91 | 0.88 | 0.93 | 1.03 |
| Psipred | PSSM | 2.25 | 4.54 | 9.11 | 18.46 | 38.14 | 78.99 | 163.26 | 343.04 | 721.06 |
| | Pred | 0.15 | 0.15 | 0.15 | 0.15 | 0.15 | 0.15 | 0.15 | 0.15 | 0.15 |
| DeepCNF | PSSM | 2.24 | 4.55 | 9.16 | 18.55 | 38.26 | 79.94 | 165.24 | 346.57 | 722.20 |
| | Pred | 16.84 | 29.45 | 54.59 | 110.76 | 207.12 | 382.16 | 625.98 | 1061.38 | 2114.31 |
| RaptorX | PSSM | 3.31 | 6.89 | 14.23 | 29.71 | 61.78 | 132.45 | 280.70 | 606.79 | 1271.95 |
| | Pred | 0.90 | 0.90 | 0.90 | 0.90 | 0.90 | 0.90 | 0.91 | 0.91 | 0.91 |
| SSpro8 | PSSM | 1.73 | 3.69 | 7.80 | 16.61 | 35.43 | 76.30 | 160.73 | 342.09 | 714.13 |
| | Pred | 4.38 | 4.67 | 4.95 | 5.19 | 5.37 | 5.55 | 5.69 | 5.86 | 5.92 |
| SSpro8+ | PSSM | 2.32 | 4.62 | 9.17 | 18.42 | 37.88 | 79.08 | 163.87 | 344.39 | 724.29 |
| | Pred | 6.15 | 6.16 | 6.14 | 6.12 | 6.12 | 6.13 | 6.13 | 6.13 | 6.14 |

The unit of these data is second. Each **PSSM** generation and **Pred**iction time was measured and averaged with ten repeats. The query set applied here was the TS115. For the results of the CASP query sets, or the results of the target datasets with <0.15 million proteins, please see S2 Table.

process. The SSpro8, for example, reduces its time cost by setting higher E-value thresholds for the PSI-BLAST in sequence similarity search. However, this adjustment does not save much time; for target sets larger than a million proteins, it only cuts down the time cost by 4.6% on average. Meanwhile, it dramatically sacrifices the accuracy (Fig 1 and S2 Table). We have proposed a simple strategy to enhance the speed of SSP without sacrificing accuracy: shrinking the target dataset by homology reduction and random sampling. We have shown that, although size reduction of the target dataset will decrease the accuracy, homology reduction will increase it. By choosing a suitable dataset size, like the recommended 5 million proteins, the effects of size and homology reduction will be neutralized. Besides, by fixing the size of the target dataset, as the UniRef up to date becomes more massive, more running time will be saved. We have demonstrated that for the UniRef90-2015 (38.2 million) and UniRef90-2018 (87.3 million) datasets, our strategy speeds up SSP by 9.3 and 20.9 folds, respectively. It is expected that, in 2020, when the UniRef90 grows to ~160 million proteins, using this strategy will enhance the SSP speed by around 40 folds. At present, using an accurate algorithm to predict the secondary structure of one protein may take nearly an hour. Considering that the time cost will grow in proportion to the size of UniRef90, the speed enhancement achieved by this strategy shall be highly valuable, especially for large-scale studies such as the functional assignment of hypothetical proteins, structural annotation of proteins determined by structural genomics projects, predicting or analyzing protein interactomes, and other post-genomics applications like our computer-aided protein engineering system (see Future works).

## On the effect of homology reduction of target dataset on the performance of SSP

Reducing the homology of the target dataset will accelerate SSP; this is mainly because the dataset is shrunk consequently. Demonstrated in Figs 3 and 5, the homology reduction itself

exerts improving but minor effects on the speed. As for the accuracy, the effects of target set homology reduction seemed minor, too; specifically, there was only a $\leq 0.7\%$ increase in either Q3 or Q8. However, it would be too hasty to conclude that the homology of the target dataset has no or little effect on accuracy. After all, the accuracies obtained in these experiments were all higher than those predicted by the functions of target dataset size, Eqs (2) and (3). Especially in the experiment of Fig 5, where the only factor that might influence the accuracy was the sequence homology, it was evident that the difference in accuracy between the 90% and 25% homology datasets was statistically significant ($p$-values $< 10^{-5}$). These small but significant improvements led us to infer that the influence of the homology of the target dataset on SSP accuracy might not be small but just difficult to detect under large dataset size. The PSSM of a query sequence is generated based on the homologs identified by PSI-BLAST in its iterative similarity search against the target dataset. During the search, PSI-BLAST, by default, only considers the top 500 hits ranked by the E-value [21]. When the dataset is enormous, no matter at what homology level we have tested, for most query proteins, the number of homologs might always be far more than 500 and make the produced PSSMs generally similar. The influence of sequence homology might thus be barely detectable. If smaller target datasets were applied, the effects of homology reduction on SSP accuracy might be more prominent, and we expected that such effects could be detected by measuring the information entropy of the PSSM. More studies shall be conducted to examine this inference.

## On the relationship between the information entropy of PSSM and the accuracy of SSP

Based on the experiments of Figs 4 and 6, we found that the Shannon entropy of PSSM is positively correlated with SSP accuracy, especially when there was only one changing factor. In Fig 6, the target dataset size was fixed while the homology was decreasing, and the correlation coefficients between the entropy and all accuracies were $\geq 0.917$. To further test whether the entropy can be applied to explain the accuracy accomplished by a PSSM, we now turn to the first experiment again. In that experiment, while the target dataset size was decreasing, the homology of datasets was fixed (90% identity)–meaning the target dataset size is the only changing factor. Since the size of the target dataset exerts a more significant influence on the accuracy than the homology does, the correlation obtained here is expected to be stronger than that of Fig 6. Indeed, as shown in Fig 7 (query: TS115) and S8 Fig (query: CASP12, CASP13), when the entropy is plotted with the accuracy, both curves similarly go down deep as the dataset shrinks. Regarding the TS115 query set, the correlation coefficient between the entropy and Q3 is 0.997, and the coefficients for Q8, SOV3, and SOV8 are all 0.996. As for the CASPs, the correlation coefficient between the entropy and any accuracy measure is $\geq 0.979$. In general, the Shannon entropy is associated with the extent of concentration of the probability distribution of a variable. If the distribution is concentrated on just a few values, the entropy is low. High entropy is obtained when the distribution is dispersed. Intuitively, a PSSM of low entropy looks simple because the observed probabilities for most amino acids are zero. In contrast, a PSSM of high entropy looks complicated (see S1 File). Hence, our data imply that 1) when the homology of the target dataset is fixed, the complexity of the PSSM will increase as the dataset size increases and, 2) when the target dataset size is fixed, the complexity of the PSSM will increase as the homology decreases. We suppose that the reason why the entropy rises as the homology reduces is that from a low-redundant target dataset, the PSI-BLAST can retrieve a more divergent set of homologs than from a highly-redundant dataset. According to the algorithm of PSSM, the more divergent the retrieved sequences, the more complicated the position propensity matrix will be constructed [21,30]. Hence, a PSSM of divergent homologs

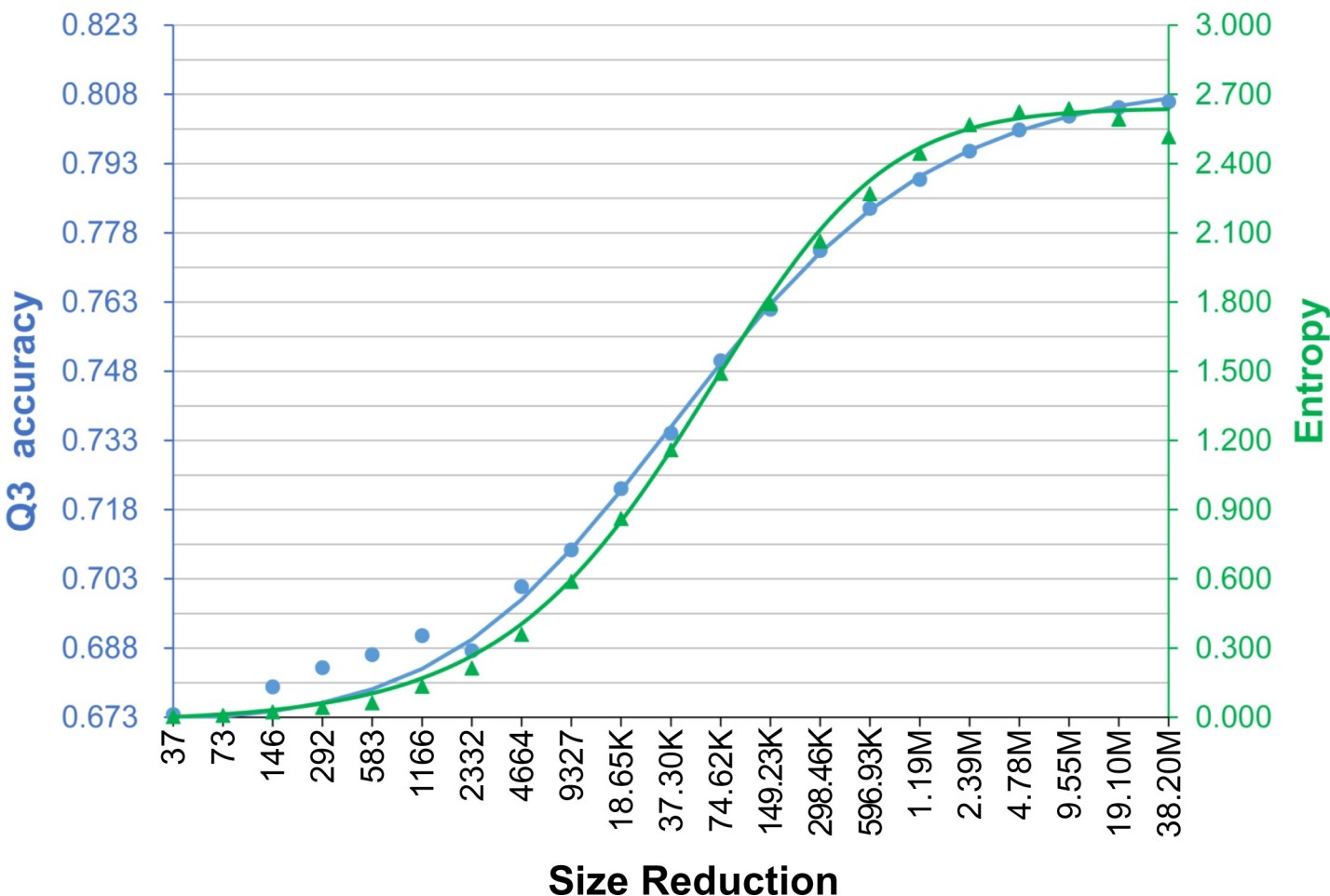

**Fig 7. The average information entropy and Q3 of the assessed secondary structure prediction methods, as the size of the target dataset reduces at a fixed homology level.** The horizontal axis indicates the size of the PSSM target datasets (M: million, K: thousand). The color codes of entropy and Q3 are the same as Fig 4. The sequence homology of the target datasets, *e.g.*, randomly-sampled subsets of the UniRef90-2015, were all the same. As the homology was fixed, the curves of the information entropy of PSSM and the accuracy of SSP showed very similar trends. The correlation coefficient between these two variables was 0.997.

will produce high entropy. Since a higher entropy stands for more encoded information, the amount of information in the SSP feature set (the PSSM) may increase and eventually facilitate machine learning and predictions, probably explaining why the accuracy is improved as the homology of the target dataset decreases.

In most experiments, we applied size and/or homology reduction to the PSSM target data-set, except for the one on the NCBI NR dataset (*i.e.*, the NrNCBI100-2020). The fact that the accuracy obtained with this vast target dataset of high homology was much lower than the accuracy obtained with UniRef90-2015 or UniRef90-2018 could be well reflected by the entropy. The average information entropy of PSSM of the TS115 query set produced with NrNCBI100-2020 was 1.836, much smaller than that produced with UniRef90-2015 (2.516) or UniRef90-2018 (2.502).

Although the information entropy seems promising to explain how the quality of a PSSM influences the accuracy of SSP, the entropy is simultaneously affected by the size and homol-ogy of the target dataset, and this complicated situation has made the correlation between the entropy and accuracy not always strong (Fig 4). In this work, we proposed a feasible way to assess the quality and SSP performance of a PSSM. Nevertheless, dissecting the detailed

mechanism underlying the relationship between information entropy and SSP accuracy remains challenging. To our knowledge, there have been similar findings reported previously. In the study by Wang *et al.* [23], a measure termed Neff was defined to quantify the quality of a PSSM, and this measure also positively correlated with SSP accuracy. Judging from the formula provided in [23], the Neff was, in fact, a derivative of the Shannon information entropy.

### Future works

We have long been interested in computer-aided protein engineering and developing novel bioengineering techniques by applying protein structural phenomena like circular permutation and three-dimensional domain swapping. Previously we have developed several structure-based algorithms to study these phenomena [37–41]. However, most known proteins have no structural data yet, but only amino acid sequences. To strengthen our bioengineering platform, we have been developing sequence-based algorithms to identify and analyze these phenomena and found secondary structure prediction a great help. Fortunately, convenient SSP algorithms have been available, inclusive of the excellent ones utilized in this study [23–28]. Thanks to these algorithms, our researches could be moved forward. Nevertheless, to test our sequence-based bioengineering platform involved a vast number of proteins, and accurate SSP predictors might take an hour to process just one (Table 1). As we tried to find solutions to enhance efficiency, we formed hypotheses, performed experiments, and eventually figured out a way to reduce the time cost significantly while preserving high accuracy. Shortly, we will apply the proposed strategy and state-of-the-art SSP methods to the sequence-based prediction of the circular permutation sites for proteins, the sequence-based detection of 3D domain swapping, and the enhancement of an SSP-driven template search algorithm for protein structure modelling [42] that can be applied to predict the structure of circularly-permuted or domain-swapped proteins.

### The database of homology reduced datasets of the UniRef

To our knowledge, there are not yet web resources providing UniRef non-redundant sets with sequence identity <50%. However, according to our experimental results, the recommended identity of the PSSM target dataset is 25%. Since most homology clustering software applicable to massive datasets (like the CD-HIT [43]) does not support identities lower than 40%, and most software capable of making non-redundant sets of low identities (such as the 32-bit USEARCH [44]) cannot process massive datasets, it may not be easy to create a homology-reduced UniRef dataset of 25% identity. Although the homology reduction procedure we described in Materials and Methods is applicable, without a machine cluster supported by an efficient distributed computation system, it may take weeks to create a 25% identity NR set from the source UniRef90. To assist researchers in applying the proposed strategy, we have established a web-based database providing NR sets of the UniRef with 25% as the lowest identity level. All the experimental datasets of this study are provided as well. This database will be updated semiannually to keep up with the growth of UniRef and is available at http://10.life.nctu.edu.tw/UniRefNR.

### Conclusions

Based on the observations that the improvements in SSP accuracy are limited regardless of the rapid growth of protein sequences in recent years, and that the amount of zero probability of a PSSM would influence SSP accuracy, we hypothesized that 1) the number of target sequences might have a greater effect on the time cost than on the accuracy of SSP, and 2) the homology of target sequences might affect the quality of generated PSSM as well as the SSP accuracy. Accordingly, it was expected that 1) size reduction of the target dataset would substantially cut

down the time cost without much degrading the accuracy, and 2) homology reduction of the target dataset would increase the complexity of the PSSM and improve SSP accuracy. Experimental data of size/homology reductions of the target dataset agreed with the expected results and thus supported our hypotheses. We also discovered that the Shannon information entropy could measure the complexity of a PSSM and might help explain the accuracy it produces. Based on these findings, we proposed two strategies to speed up SSP without sacrificing accuracy or even with enhancements: 1) a homology reduction of the target dataset accompanied by a size reduction to millions of proteins, or 2) merely a homology reduction of the target dataset. Tested with the UniRef of 2018, the first strategy reduced the average time cost of state-of-the-art SSP algorithms from 40.6 to 1.9 min with a <0.7% decrease in Q3 or Q8 accuracy. As for the second strategy, the time cost was reduced to 48.4% while the accuracy was increased up to 0.8%. This study proves that SSP applications do not need to keep using the huge UniRef90 target dataset, which is exponentially growing and extremely challenging the computing power and storage capacity of researchers. For large-scale post-genomics applications of SSP, using the proposed strategy not only will save much time but may increase the reliability of data because of the improvement in SSP accuracy.

## Materials and methods

### Experimental environments

This study was carried out using three server machines. All experiments were performed on one machine equipped with two hyperthreading 6-core (Intel Xeon) 3.33 GHz processors and 166 GB memory. The other two possessed two hyperthreading 4-core (Intel Xeon) 2.27 GHz processors and four 12-core (AMD Opteron) 2.20 GHz processors, respectively. They were all clustered with a distributed computation system we developed [41,45] to speed up the production of homology-reduced UniRef datasets and sustain the periodic updating of the database constructed in this work.

### Experimental datasets

For assessing the performance of secondary structure prediction methods, a set of query sequences and a target dataset for generating PSSM are required. Since the state-of-the-art SSP methods assessed in this study were all developed before 2016, meaning their predictive models were all trained with proteins released at the PDB no later than Dec. 2015. Hence, an ideal query dataset for evaluating prediction accuracy should be composed of proteins released after Jan. 1st 2016. In a review by Dr. Yaoqi Zhou [17], an independent test query dataset TS115 (115 proteins) constituted with proteins of PDB 2016 had been established and was, therefore, a reasonable choice for this study. According to [17], the sequence identity between TS115 and the protein structures released in PDB before 2016 was ≤30%. Besides, the CASP12 (46 proteins) and CASP13 (43 proteins) datasets obtained from the 12th and 13th biannual meeting of Critical Assessment of Structure Prediction techniques [46], which also comprised novel protein structures determined after Jan. 2016, were used as the query sets. As for the source of target sequences, the UniRef90 of the year 2015 (UniRef90-2015; 38.2 million proteins) and 2018 (UniRef90-2018; 87.3 million proteins) established by the UniProt [47], and the non-redundant protein dataset prepared by the NCBI in 2020 [36] (NrNCBI100-2020; 257.1 million proteins), were utilized.

### Applied secondary structure prediction algorithms

Several SSP algorithms were applied to perform the experiments. The standalone programs of them were all released before 2016, including three-state algorithms: Psipred (v3.3) [22],

SpineX (v2.0) [24], Scorpion (v1.0) [26], and Spider2 (v2.0) [27], and eight-state algorithms: RaptorX (v1.0) [23], SSpro8 (v5.2) [25], and DeepCNF (v1.02) [28]. The original packages of these programs recruited different versions of PSI-BLAST as the PSSM generator. In order to make fair assessments, we modified their scripts to uniformly use the psiblast program of NCBI blast 2.6.0 [21]. For each algorithm, the parameters of PSI-BLAST were set according to their original scripts, except for the SSpro8 (see Results). The parameter settings of these algorithms are available in S1 Table.

## Reduction of target dataset size by random sampling

Size-reduced target datasets were generated based on the UniRef90 by random sampling. Because an experiment was repeated 10 times for each test group, when the size of a shrunk target dataset was smaller than 1/10 of the UniRef90, the sampling could be done perfectly without replacement, that is, the 10 subsets would not share any common entry. When the size was larger than 1/10, the inter-subset replacement was inevitable such that some entries might be included in more than one subset. We had tested random sampling with or without inter-subset replacements, and the results showed no statistically significant difference. Therefore, the randomly-sampled target datasets used in this study were prepared by allowing inter-subset replacements.

## Reduction of target dataset homology

The UniRef only provided three sequence identity non-redundant (NR) datasets, the UniRef100, UniRef90, and UniRef50. To finely test how the homology reduction would influence the performance of SSP algorithms, we had to make NR sets ranging from 90%, 80% to 25% identities. For homology levels ≥40% identity, the CD-HIT [43] was applied. Because CD-HIT did not support lower identities, for homology levels <40% identity, we changed to the USEARCH [44]. However, the 32-bit USEARCH had a limit of 4 GB memory usage, and the 40% identity NR set generated by CD-HIT was still too huge to apply the USEARCH directly. We hence designed the following procedure (illustrated in Fig 8) with USEARCH and BLAST as the homology reduction engines to generate low identity NR sets from a source dataset containing a vast number of sequences,

1. Sort the sequences of the input dataset by the length in descending order.

2. Divide the sorted sequences into subsets, each containing $m$ proteins. Let $S_1, S_2 \ldots, S_N$ denote the produced subsets, where $N$ is the number of subsets.

3. Intra-subset homology reduction.

    1. Let $h$ denote the homology level of the final output dataset in percent.

    2. For each subset, apply USEARCH to generate its $h$ % sequence identity NR set and replace the original subset by this NR set.

    3. After all the subsets are replaced by the NR sets, within every subset, any two sequences will share $< h$ % sequence identity.

4. Inter-subset homology reduction.

    1. Let $x = 1$ and take subset $S_x$ to be the head set, or, the invariable set.

    2. From the subsets other than the head set, choose $S_y$, where $y \leq N$, to be the body set, or, the variable set.

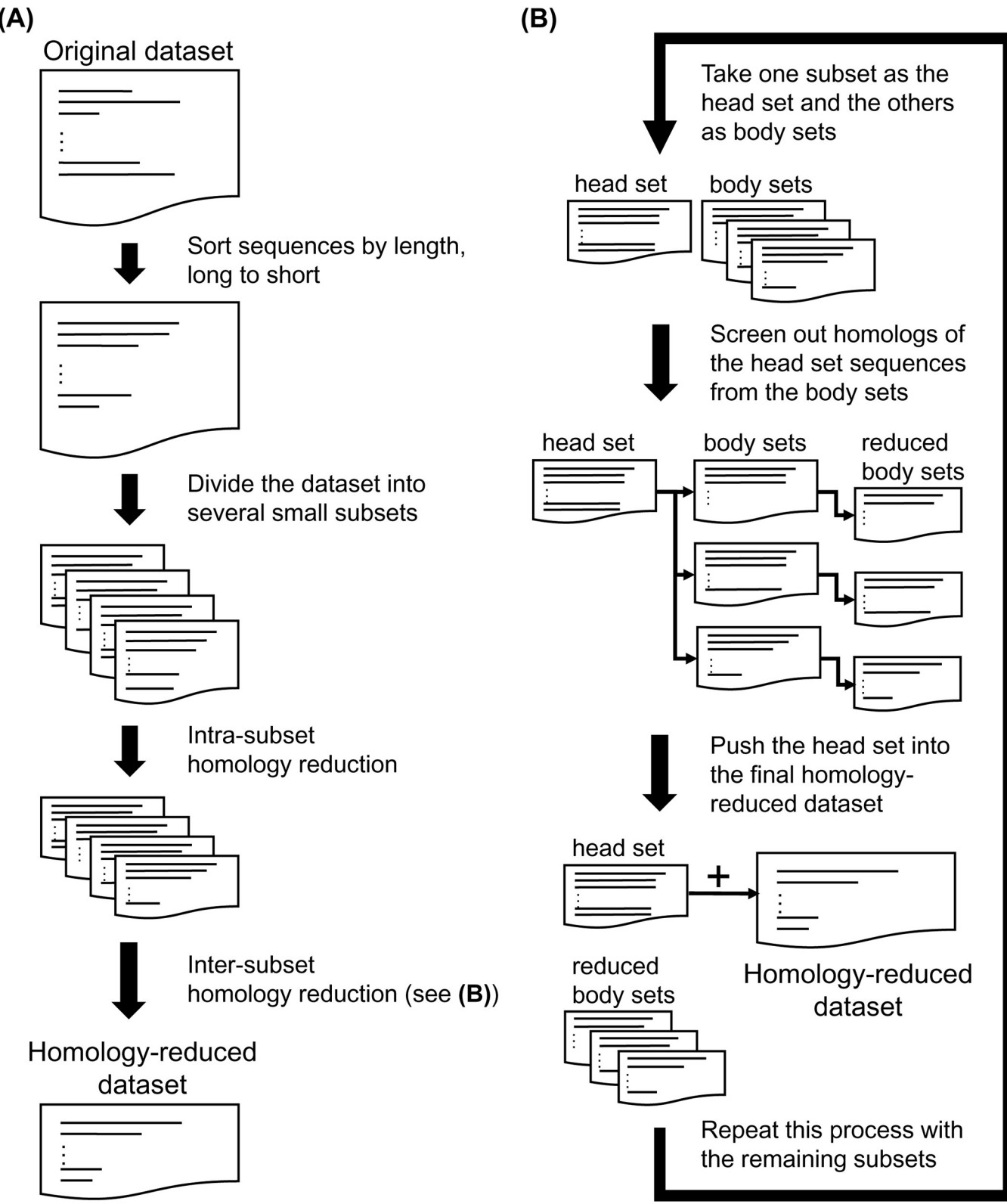

**Fig 8. The procedure of sequence homology reduction of the target dataset.** The sequence identity non-redundant subsets of the UniRef90 was created using this procedure, which first divides the original dataset into subsets and performs a round of intra-subset homology reduction for each subset and iterative rounds of inter-subset homology reduction for every pair of subsets. The sequence homology reduction was made using the USEARCH and BLAST.

3. Merge the head set and body set, run USEARCH to screen out homologs with identities $\geq h$ %. Reassemble the remaining sequences back into the head and body sets. Let $S_x{'}$ and $S_y{'}$ denote the reduced version of $S_x$ and $S_y$, respectively.

4. Let $q$ denote a sequence removed from $S_x$ by USEARCH, that is, $q \in S_x - S_x{'}$. Using BLAST, take $q$ as the query sequence, find its homologous sequences with identities $\geq h$ % from $S_y{'}$, and then eliminate those sequences from $S_y{'}$. Repeat this step until the homologs of all $q$ are eliminated from $S_y{'}$.

5. Let $S_y$ be replaced by the last $S_y{'}$. Now the remaining sequences in $S_y{'}$ all share $< h$ % identities with the sequences in $S_x$.

6. Repeat steps (2) to (5) using another subset to be the body set until no more subset can be applied. Then, push the sequences of $S_x$ into the collection of NR sequences and delete $S_x$.

7. Repeat steps (1) to (6) by setting $x = x + 1$ until no subset is available to be the body set (*i.e.*, $x = N$). Push the final head set $S_N$ into the collection of NR sequences.

5. Save the collection of NR sequences as the final output dataset, in which any two proteins share $< h$ % sequence identity.

In this procedure, the head set $S_x$ is also termed the invariable set because, during the inter-subset homology reduction step, it remains unchanged; homologous sequences are removed from the body sets, which are thus termed the variable sets. The size $m$ of the divided subsets is subject to the limit of memory usage of the USEARCH; in this work, it was set to be 100,000 proteins.

## Statistical analysis

The tests performed with random sampling in this study were all repeated ten times. For each test group, the time cost and accuracy measures were averaged, and the sample standard deviation of those measures was calculated. For checking whether the observed difference between two groups was statistically significant, several tests were made to compute the *p*-value. First, the Shapiro-Wilk test was applied to check the normality of the measure values of each group. Next, if the normal distribution of the values was verified, the *F*-test was performed to determine the equality of variances for the two groups. Finally, if the two groups were verified to come from populations with equal variance by the *F*-test, we used the Student's *t*-test to compute the *p*-value; otherwise, the Welch's *t*-test was applied.

## Accuracy measures

**The Q and SOV measures.**   In addition to the traditional Q3 and Q8 accuracy, we also evaluated SSP methods by the SOV (segment overlap) measure [48,49]. The definition of the Q accuracy was the number of correctly predicted residues divided by the number of predicted residues. The difference between Q3 and Q8 was the type of secondary structural element (SSE) codes applied in the prediction, *i.e.*, three-state or eight-state codes. The SOV is a measure for evaluating the accuracy of SSP based on secondary structure segments instead of residues. It is generally regarded as a more critical way of assessment than the conventional Q, for its capability of capturing the overall quality of SSP for a protein and reducing noises from individual residues [48–50]. Previous works calculated the SOV based on the three-state SSE, or, the SOV3. In this study, the eight-state SOV (SOV8) was also calculated.

**The macro-average, micro-average, and weighted average of accuracy measures.** In general, when assessing an SSP algorithm, multiple query proteins are used. Therefore, the presented accuracy value is usually an average. In most previous SSP researches, an accuracy value was computed by averaging the accuracies of all individual query proteins using the classic arithmetic mean equation $sum/n$, where $sum$ is the summation of accuracy from all proteins and $n$ is the number of proteins. However, this classic average weighs every protein in the query dataset equally, which is questionable because the size of query proteins may be very different. For example, the size of the TS115 proteins ranges from 43 to 1,085 residues.

In this study, the Q3 and Q8 were computed based on residues instead of proteins, *i.e.*, the total number of correctly predicted residues from all query proteins divided by the total number of residues from all query proteins. The SOV3 and SOV8 measures could not be computed based on residues and were thus averaged over proteins with protein size as the weight, that is,

$$S\bar{O}V = \frac{\sum_{i=1}^{n} size_i \times SOV_i}{\sum_{i=1}^{n} size_i} \tag{9}$$

where $S\bar{O}V$ represents the weighted average of SOV, $n$ denotes the number of query proteins, $size_i$ and $SOV_i$ stand for the size (number of residues) and SOV value of protein $i$, respectively.

Computing the averaged accuracy based on proteins and residues is analogous to computing the macro-average and micro-average of the accuracy values, respectively. Both the micro-average Q3/Q8 and size-weighted average SOV3/SOV8 prevent underestimating the influence of large proteins or overestimating that of small ones, and hence they can precisely reflect the actual accuracy of SSP methods.

## Computation of the information entropy of a PSSM

The information entropy proposed by Shannon measures the amount of information in a variable. It is also known as the disorder or uncertainty of a set of data [35]. The Shannon entropy $S$ in the case of a multi-value or multi-state variable is given by the following formula,

$$S = -\sum p_c log_2(p_c) \tag{10}$$

where $c$ stands for a specific value or state of the variable, and $p_c$ is the observed probability of $c$ in the entire probability distribution of the variable.

A traditional PSSM for a residue position contains 20 values, each assigned to an amino acid. The meaning of those values is, based on the multiple alignment between the query sequence and a set of identified homologous sequences, how possible in evolution the residue position of interest could be substituted with each of the 20 amino acids. For a highly conserved residue position, the probability distribution of the 20 amino acids is usually simple or concentrated; oppositely, for an evolutionarily diverse position, the distribution is complex and dispersed. Since the PSSM is by nature the probability distribution of a 20-state variable, we supposed that the Shannon entropy could be applied to quantify the complexity of a PSSM. Because for each residue, the PSSM output of PSI-BLAST readily provided the observed probabilities in percentage, the information entropy could be directly computed using Eq (10).

## Supporting information

**S1 File. Example of PSSMs generated from homologs with high or low sequence homology.** (PDF)

**S2 File. The time cost and Q3 accuracy of the assessed SSP methods as the length of target sequences increases.**
(XLSX)

**S1 Fig. Detailed results of the size reduction of the PSSM target dataset on three independent query sets.**
(TIF)

**S2 Fig. The time cost and accuracy of individual SSP methods assessed as the sequence homology of the target dataset reduces.**
(TIF)

**S3 Fig. The average time cost and accuracy of the assessed SSP methods on the CASP query sets as the sequence homology of the target dataset reduces.**
(TIF)

**S4 Fig. The average information entropy and Q3 accuracy of the assessed SSP methods on the CASP sets as the sequence homology of the target dataset reduces.**
(TIF)

**S5 Fig. The time cost and accuracy of individual SSP methods assessed as the homology of the target dataset reduces with a fixed dataset size.**
(TIF)

**S6 Fig. The average time cost and accuracy of the assessed SSP methods on the CASP query sets, as the sequence homology of the target dataset reduces with a fixed dataset size.**
(TIF)

**S7 Fig. The average information entropy and Q3 of the assessed SSP methods on the CASP query sets, as the homology of the target dataset reduces with a fixed dataset size.**
(TIF)

**S8 Fig. The average information entropy and Q3 of the assessed SSP methods on the CASP query sets, as the size of the target dataset reduces at a fixed homology level.**
(TIF)

**S1 Table. PSI-BLAST settings of the assessed SSP methods.**
(XLSX)

**S2 Table. The average and individual performance data of state-of-the-art SSP methods assessed with size-reduced PSSM target datasets.**
(XLSX)

**S3 Table. The average and individual performance data of state-of-the-art SSP methods assessed with homology-reduced PSSM target datasets.**
(XLSX)

**S4 Table. The average and individual performance data of state-of-the-art SSP methods assessed with size- and homology-reduced PSSM target datasets.**
(XLSX)

**S5 Table. The average and individual performance data of state-of-the-art SSP methods assessed with the UniRef sequences of 2018.**
(XLSX)

**S6 Table. Results of curve fitting for the average time cost and accuracy of the SSP methods assessed with target datasets of decreasing homology.**
(XLSX)

**S7 Table. The time cost and accuracy of state-of-the-art SSP methods on the NrNCBI100-2020 target dataset.**
(XLSX)

## Acknowledgments

We would like to thank Chia-Hua Lo, a student of WCL, for editing and proofreading the manuscript. The progress of this study was greatly accelerated owing to the computing power offered by Prof. Jinn-Moon Yang at National Chiao Tung University, Hsinchu, Prof. Jenn-Kang Hwang at the Chinese University of Hong Kong, Shenzhen, and Prof. Ping-Chiang Lyu at National Tsing Hua University, Taiwan.

## Author Contributions

**Conceptualization:** Wei-Cheng Lo.

**Data curation:** Sheng-Hung Juan, Teng-Ruei Chen, Wei-Cheng Lo.

**Formal analysis:** Sheng-Hung Juan, Wei-Cheng Lo.

**Funding acquisition:** Wei-Cheng Lo.

**Investigation:** Sheng-Hung Juan, Teng-Ruei Chen, Wei-Cheng Lo.

**Methodology:** Sheng-Hung Juan, Wei-Cheng Lo.

**Project administration:** Wei-Cheng Lo.

**Resources:** Teng-Ruei Chen, Wei-Cheng Lo.

**Software:** Sheng-Hung Juan, Teng-Ruei Chen, Wei-Cheng Lo.

**Supervision:** Wei-Cheng Lo.

**Validation:** Sheng-Hung Juan, Teng-Ruei Chen, Wei-Cheng Lo.

**Writing – original draft:** Sheng-Hung Juan, Teng-Ruei Chen, Wei-Cheng Lo.

**Writing – review & editing:** Sheng-Hung Juan, Wei-Cheng Lo.

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
