## [Decision Letter · Decision Letter 0]

6 Mar 2020

PONE-D-19-35005

A simple strategy to enhance the speed of protein secondary structure prediction without sacrificing accuracy

PLOS ONE

Dear Prof. Lo,

Thank you for submitting your manuscript to PLOS ONE. After careful consideration, we feel that it has merit but does not fully meet PLOS ONE’s publication criteria as it currently stands. Therefore, we invite you to submit a revised version of the manuscript that addresses the points raised during the review process.

We would appreciate receiving your revised manuscript by Apr 20 2020 11:59PM. To enhance the reproducibility of your results, we recommend that if applicable you deposit your laboratory protocols in protocols.io, where a protocol can be assigned its own identifier (DOI) such that it can be cited independently in the future. For instructions see: http://journals.plos.org/plosone/s/submission-guidelines#loc-laboratory-protocols

We look forward to receiving your revised manuscript.

Kind regards,

M. Sohel Rahman, Ph.D.

Academic Editor

PLOS ONE

Additional Editor Comments (if provided):

Please attend to all the comments of the reviewers and carefully revise the manuscript. If you decide not to follow some comments, you MUST provide an appropriate rebuttal.

Journal Requirements:

Reviewers' comments:

Reviewer's Responses to Questions

**Comments to the Author**

1. Is the manuscript technically sound, and do the data support the conclusions?

Reviewer #1: Yes

Reviewer #2: Yes

2. Has the statistical analysis been performed appropriately and rigorously? 

Reviewer #1: Yes

Reviewer #2: N/A

3. Have the authors made all data underlying the findings in their manuscript fully available?

Reviewer #1: Yes

Reviewer #2: Yes

4. Is the manuscript presented in an intelligible fashion and written in standard English?

Reviewer #1: Yes

Reviewer #2: No

5. Review Comments to the Author

Reviewer #1: In this paper, the authors have managed to prove their proposed hypothesis about the reduction of prediction time of secondary structure prediction. The reduction of target data set size as well as homology reduction of the target sequence have managed to cut the prediction tie by a significant margin. However, the following points should be addressed for further enhancement of the article:

1) The hypothesis was tested entirely on the Uniref90 data set, hence the equations (1), (2) and (3) of Tc, Q3 and Q8 respectively are based on this data set. But are these equations universal? Will they always follow a linear equation or other higher degree polynomials? Such questions might be answered by validating the hypothesis on another recent data set, for example CASP12 or CASP13.

2) The average information content or entropy of a PSSM for a particular target was measured by Shannon's equation which uses base 2 because of it's original implementation of measuring information in bit stream. Although we can convert between any base, the reasons for using base "e" was not cleared in manuscript.

3) Does the impact of reduction of target data size on accuracy depend on the length of the target sequence? An analysis based on different lengths of targets can be helpful here.

Overall the paper is written in well structured English and organized in sequential step by step points proving the results of the hypothesis.

Reviewer #2: This paper presents an interesting piece of work where authors described two simple strategies to reduce the time cost of a classical problem in the field of sequence-based prediction of structural properties of proteins, namely the secondary structure prediction (SSP).

1. The authors highlighted that computation of PSI-BLAST is the major time-consuming element of the SSP due to it’s prerequisite to search through a target database, such as Uniref90. As a potential strategy to reduce the time cost of SSP, the authors then proposed down sampling of the target dataset to search through for PSSM generations. Using seven SSP algorithms, authors showed that the reduction of target dataset has greater effect in decreasing the time-cost (3355 fold) with maximum of 14.1% compromise in accuracy.

a. Do all the seven predictors recruited here to benchmark use UniRef90 for the blast search during PSSM computation?

b. Can authors comment on the utility of NCBI Non-redundant database available at ftp://ftp.ncbi.nlm.nih.gov/blast/db/?

c. There are SSP software that use NCBI Non-redundant database for PSSM computation. How do the time-costs required for PSSM computation using UniRef90 and NCBI NR databases compare?

d. On which dataset the predictors were run to generate Fig. 1?

2. The authors discussed the effects of sequence redundancy of the reference dataset (here, UniRef90), which I appreciate.

a. However, I would again like to see a comparison between using UniRef NR datasets and NCBI NR dataset.

b. The results section of the paper has statements, like “use of 25% identity NR datasets ‘truly’ outperformed the 80% identify dataset”. I recommend stating the actual percentage increase in accuracy in these cases.

c. As the improvement in accuracy by homology reduction did not reach a plateau, what was the rationale for stopping at 25%? Would be interesting to further reduce and check the improvement in accuracy.

d. I recommend benchmarking the accuracies using an independent set of proteins and predicting the SS using UniRef NR90 to NR25.

3. I appreciate the authors’ effort to make the non-redundant UniRef datasets publicly accessible to foster downstream utility.

4. The paper needs a thorough proof-read for English, and especially, to eliminate redundancies. The current paper is unnecessarily long with many repetitive segments.

5. All figures are of very poor quality. Figures of appropriate resolutions are highly recommended.

6. PLOS authors have the option to publish the peer review history of their article (what does this mean?). If published, this will include your full peer review and any attached files.

Reviewer #1: No

Reviewer #2: No

---

## [Author Response · Author response to Decision Letter 0]

28 Apr 2020

( A Word file of our responses to Reviewers' Comments has been uploaded. Although we make a plain text version here following the instruction of the submission system of PLOS ONE, we would like to suggest viewing the Word version. )

Dear Professors and Editor,

We would like to thank the anonymous reviewers for their careful reading and detailed comments, which have greatly enhanced this article. We are pleased that you find sufficient merit in the work as to ask for an appropriately revised manuscript. The new version of our manuscript has been modified according to referees' comments, which are also answered as follows.

Comments to the Author

1. Is the manuscript technically sound, and do the data support the conclusions?

Reviewer #1: Yes

Reviewer #2: Yes

Response:

We would like to thank the reviewers for their positive comments.

2. Has the statistical analysis been performed appropriately and rigorously?

Reviewer #1: Yes

Reviewer #2: N/A

Response:

Indeed, complicated statistical analyses are not required to conduct this research. Nevertheless, because we have done one thing very different from most previous SSP studies, i.e., multiple repeats of the experiments by random sampling, we were able to compute the standard deviation of the performance measures and the p-value between test groups. In the original draft, we had described how we compute the p-value in the subsection "Multiple repeats of experiments and statistical analysis." Considering that this title is out of the focus, we have changed it to "Statistical analysis" (Page 42) following the Submission Guidelines of PLOS ONE. We hope that this modification will make the readers more comfortable to find out how we made the significance tests.

3. Have the authors made all data underlying the findings in their manuscript fully available?

Reviewer #1: Yes

Reviewer #2: Yes

Response:

We thank the reviewers for their time and patience in reviewing our numerous supporting data files.

4. Is the manuscript presented in an intelligible fashion and written in standard English?

Reviewer #1: Yes

Reviewer #2: No

Response:

We thank the reviewers for the comments. Please see below for our response to the English writing of the revised manuscript.

5. Review Comments to the Author

Reviewer #1: In this paper, the authors have managed to prove their proposed hypothesis about the reduction of prediction time of secondary structure prediction. The reduction of target data set size as well as homology reduction of the target sequence have managed to cut the prediction tie by a significant margin. However, the following points should be addressed for further enhancement of the article:

1) The hypothesis was tested entirely on the Uniref90 data set, hence the equations (1), (2) and (3) of Tc, Q3 and Q8 respectively are based on this data set. But are these equations universal? Will they always follow a linear equation or other higher degree polynomials? Such questions might be answered by validating the hypothesis on another recent data set, for example CASP12 or CASP13.

Response:

We thank the reviewer for the positive feedback and constructive comments, which greatly help us improve this work. The trends revealed by the equations of TC, Q3 and Q8 may be universal, but the fitted constants may not. Those constants are supposed to vary according to the datasets and hardware/software used to perform secondary structure prediction (SSP). First, we have added a note after those equations on Page 12 to make the reader aware of this. Second, to verify whether the trends of those equations are universal, we have utilized both the CASP12 and CASP13 to repeat all experiments. Comparing Eq (1) and the new Eqs (4) and (5) obtained with the CASPs, the linear relationship between the SSP time cost and the target dataset size can be confirmed. To help visualize the linear relationship, we have made a plot in the new S1 Fig showing the straight lines drawn based on the results of those independent query sets. As for the accuracy, the relationship between Q3/Q8 and the target dataset size is sigmoid, no matter being tested with the original query set TS115 or the CASPs (also shown in S1 Fig). We feel so grateful for having followed the suggestion from the reviewer and used these CASP sets because all the results obtained with them well support the conclusions of this study. In the revised manuscript, we have marked the new contents about the CASPs with an orange text background color.

2) The average information content or entropy of a PSSM for a particular target was measured by Shannon's equation which uses base 2 because of it's original implementation of measuring information in bit stream. Although we can convert between any base, the reasons for using base "e" was not cleared in manuscript.

Response:

Thank you very much for indicating this difference. Actually, at the last minute before our submission of the first draft, we had noticed it and thus put down that statement. The reason why we used base "e" was just because of the default of the log() function of the programming language. We have recalculated those values with base 2 and made relevant revisions, inclusive of Eq (10). Please see Figs 4, 6, and 7, as well as the new S4, S7, and S8 Figs for the updated data. By changing to base 2, the original conclusions were not affected.

3) Does the impact of reduction of target data size on accuracy depend on the length of the target sequence? An analysis based on different lengths of targets can be helpful here.

Response:

We appreciate this interesting comment and have accordingly done that. Before this experiment, it can be reasonably expected that the time cost will grow as the length of the target sequences increases. As for the accuracy, because we have a recent study on the limit of SSP accuracy showing that current SSP algorithms perform better for long query proteins than for short ones, by extending this concept it may be expected that the accuracy of SSP will rise as the length of target sequences increases. Finally, the experimental results shown in S2 File agree with the expected. The accuracy is improved as the length of target sequences increases, but the SSP time cost is also raised. Since this article is about the efficiency improvement of SSP, we find these results do not well fit the purpose of it. Therefore, we have not yet integrated this experiment into the revised manuscript. However, if the reviewer feels it necessary to add this part to enhance the completeness of the report, we will be happy to do that.

Overall the paper is written in well structured English and organized in sequential step by step points proving the results of the hypothesis.

Response:

Thank you for your recognition of our writing quality. To improve the readability of this report, we have still made a lot of English writing optimization and deletions of redundant statements.

Reviewer #2: This paper presents an interesting piece of work where authors described two simple strategies to reduce the time cost of a classical problem in the field of sequence-based prediction of structural properties of proteins, namely the secondary structure prediction (SSP).

1. The authors highlighted that computation of PSI-BLAST is the major time-consuming element of the SSP due to it's prerequisite to search through a target database, such as Uniref90. As a potential strategy to reduce the time cost of SSP, the authors then proposed down sampling of the target dataset to search through for PSSM generations. Using seven SSP algorithms, authors showed that the reduction of target dataset has greater effect in decreasing the time-cost (3355 fold) with maximum of 14.1% compromise in accuracy.

a. Do all the seven predictors recruited here to benchmark use UniRef90 for the blast search during PSSM computation?

b. Can authors comment on the utility of NCBI Non-redundant database available at ftp://ftp.ncbi.nlm.nih.gov/blast/db/?

c. There are SSP software that use NCBI Non-redundant database for PSSM computation. How do the time-costs required for PSSM computation using UniRef90 and NCBI NR databases compare?

d. On which dataset the predictors were run to generate Fig. 1?

Response:

We would like to thank the reviewer for the questions and valuable information, which provides more materials for us to test the feasibility of the proposed strategy. The questions are answered as follows,

a:

Not exactly. The UniRef90 is commonly used, but other non-redundant sequence datasets are also utilized. For instance, Spider2 uses the NR90 dataset established by HHsuite, and Psipred uses the NCBI NR dataset. As for DeepCNF, SpineX, and SSpro8, the UniRef was applied. By referring to the system Dr. Yaoqi Zhou benchmarked state-of-the-art SSP algorithms in the review paper [32], we decided to use the UniRef database. We have mentioned in the revised manuscript on Page 23, Lines 430–431 that the Psipred uses the NCBI NR set as the PSSM target dataset.

b, c:

We thank the reviewer for reminding us to use the NCBI NR for evaluating the proposed strategy. The core of our strategy is homology reduction combined with a size reduction. The sequence homology level of the NCBI NR dataset we obtained from the provided FTP site was 100% identity, higher than the ones we had tested (90% to 25%). Besides, the size of the NCBI NR dataset was much larger than the UniRef90-2015 and UniRef90-2018 we used to evaluate the strategy. This NCBI NR thus becomes an excellent example to test the feasibility of the proposed strategy in the opposite direction. It is also valuable for verifying the results we obtained with the UniRef datasets. We have used the results of UniRef homology reduction to establish a model (the new Eqs(6)–(8)) to predict the time cost and SSP accuracy if the NCBI NR were applied as the PSSM target dataset. According to the model, the time cost of using NCBI NR is expected to be 5,571 sec, and the accuracy will be lower than that obtained with the UniRef90. As expected, the actual time cost is 5,756 sec, and the accuracy was lower than that of the UniRef90-2015 and UniRef90-2018 (Pages 23–25). Besides, the difference in accuracy between NCBI NR and UniRef90s can be well explained by the information entropy of PSSM (Page 32). In the revised article, we have marked the new contents about the NCBI NR with a light green text background color.

d:

Thank you for pointing out this omission. In the previous manuscript, since the only query set was the TS115, we mentioned it just once in the Materials and Methods. Because three independent query sets (TS115, CASP12, and CASP13) and three sources of target sequences (UniRef90-2015, UniRef90-2018, and the NrNCBI100-2020) are used in the revised manuscript, we have stated explicitly for each experiment what query and target sets were applied.

2. The authors discussed the effects of sequence redundancy of the reference dataset (here, UniRef90), which I appreciate.

a. However, I would again like to see a comparison between using UniRef NR datasets and NCBI NR dataset.

b. The results section of the paper has statements, like "use of 25% identity NR datasets 'truly' outperformed the 80% identify dataset". I recommend stating the actual percentage increase in accuracy in these cases.

c. As the improvement in accuracy by homology reduction did not reach a plateau, what was the rationale for stopping at 25%? Would be interesting to further reduce and check the improvement in accuracy.

d. I recommend benchmarking the accuracies using an independent set of proteins and predicting the SS using UniRef NR90 to NR25.

Response:

We thank the reviewer for the suggestions. The questions are answered as follows,

a:

We appreciate the reviewer's suggestion about using the NCBI NR, which has helped us examine the proposed strategy more thoroughly than we did in the previous manuscript. Please see our response to your question 1b and 1c for the comparison between UniRef and NCBI NR datasets.

b:

Thank you for the constructive comment. In the previous manuscript, immediately after most summary statements we put supporting sentences. For instance, after the one you mentioned, we said: "Although the average differences in Q3 and Q8 between the 90% and 25% NR datasets were both only 0.70%, the average p-values were <1.25×10–6 and <9.99×10–8, respectively, …" In the revised manuscript, we have removed many unnecessary adverbs, adjectives, and summary statements (including the one we are talking about on Page 18). Hopefully, this "redundancy reduction" has improved the quality of our manuscript.

c:

In the experiment of Fig 2, we found that 5 million proteins would be the suitable target dataset size for accelerating SSP without sacrificing accuracy. In the experiments of homology reduction, the lowest sequence identity was 25% because when a lower identity was applied, the number of sequences remaining in the target dataset would be smaller than 5 million (see the new S6 Table) and cause unreasonable decreases in accuracy because of the small dataset size. We have added this explanation to the legend of Figs 2 and 3. In fact, if from the beginning a smaller size were applicable, the accuracy might keep increasing as the homology of the target dataset was reduced to lower than 25%. In one of our recent studies on SSP (submitted to PLOS ONE), tiny target sets of 10,000 proteins were prepared to dissect the influence of dataset sequence redundancy on the evaluation of SSP methods. In that study, we discovered that the accuracy of SSP would keep increasing down to a 10% identity homology reduction of the target set.

d:

We would like to thank the reviewer for this great quality-improving suggestion. We have applied the CASP12 and CASP13 independent sets to repeat all experiments, and the results agree well with our original conclusions. In the revised article, we have marked the new contents about the CASP tests with an orange text background color.

3. I appreciate the authors' effort to make the non-redundant UniRef datasets publicly accessible to foster downstream utility.

Response:

Thank you for the kind comment and your recognition of our effort.

4. The paper needs a thorough proof-read for English, and especially, to eliminate redundancies. The current paper is unnecessarily long with many repetitive segments.

Response:

We would like to thank the reviewer for pointing out these shortages. As non-native English speakers, we spent a lot of time writing. To improve the quality of this article, we have re-examined it and corrected several grammar or spelling errors. As for the redundancies, it was partly because we took PLOS ONE's suggestion of the figure legend too seriously: "allow readers to understand it (the figure) without referring to the text." We have removed many repetitive statements from the figure legends and simplified the main text, hoping to improve the reader's reading experience. The tracking of Microsoft Word was retained in the uploaded "Revised Manuscript with Track Changes" file. We hope these changes will meet your expectations.

5. All figures are of very poor quality. Figures of appropriate resolutions are highly recommended.

Response:

We thank the reviewer for reminding this. The figures embedded in the manuscript PDF file created by the submission system of PLOS ONE were all resampled. The original high-resolution figures can be obtained through the "Click here to access/download" hyperlink at the top of each figure page. The original figures were prepared according to the resolution required by the PLOS ONE, and the width was 2,250 pixels. If the reader uses a 100% scale to view a figure, all the points, curves, and text will be very sharp. However, we put eight panels in some figures. The points and curves on them are indeed too small for readers who do not use the viewing software that is convenient for zooming. We have modified those figures (inclusive of supporting figures) by magnifying the points and text.

6. Do you want your identity to be public for this peer review?

Reviewer #1: No

Reviewer #2: No

Response:

We would like to express again our sincere thanks to the anonymous reviewers for helping us improve this work.

---

## [Decision Letter · Decision Letter 1]

10 Jun 2020

A simple strategy to enhance the speed of protein secondary structure prediction without sacrificing accuracy

PONE-D-19-35005R1

Dear Dr. Lo,

We’re pleased to inform you that your manuscript has been judged scientifically suitable for publication and will be formally accepted for publication once it meets all outstanding technical requirements.

Kind regards,

M. Sohel Rahman, Ph.D.

Academic Editor

PLOS ONE

Additional Editor Comments (optional):

Reviewers' comments:

Reviewer's Responses to Questions

**Comments to the Author**

1. If the authors have adequately addressed your comments raised in a previous round of review and you feel that this manuscript is now acceptable for publication, you may indicate that here to bypass the “Comments to the Author” section, enter your conflict of interest statement in the “Confidential to Editor” section, and submit your "Accept" recommendation.

Reviewer #1: All comments have been addressed

Reviewer #2: All comments have been addressed

2. Is the manuscript technically sound, and do the data support the conclusions?

Reviewer #1: (No Response)

Reviewer #2: Yes

3. Has the statistical analysis been performed appropriately and rigorously? 

Reviewer #1: (No Response)

Reviewer #2: Yes

4. Have the authors made all data underlying the findings in their manuscript fully available?

Reviewer #1: (No Response)

Reviewer #2: Yes

5. Is the manuscript presented in an intelligible fashion and written in standard English?

Reviewer #1: (No Response)

Reviewer #2: Yes

6. Review Comments to the Author

Reviewer #1: (No Response)

Reviewer #2: All comments have been addressed. Especially, the newly added results reporting on the evaluation of the proposed technique on the NR dataset in the revised paper is compelling.

7. PLOS authors have the option to publish the peer review history of their article (what does this mean?). If published, this will include your full peer review and any attached files.

Reviewer #1: No

Reviewer #2: No

---

## [Editor Report · Acceptance letter]

19 Jun 2020

PONE-D-19-35005R1 

A simple strategy to enhance the speed of protein secondary structure prediction without sacrificing accuracy 

Dear Dr. Lo:

I'm pleased to inform you that your manuscript has been deemed suitable for publication in PLOS ONE. Congratulations! Your manuscript is now with our production department. 

Kind regards, 

on behalf of

Dr. M. Sohel Rahman 

Academic Editor

PLOS ONE